# The Stage-Based Model of Addiction—Using *Drosophila* to Investigate Alcohol and Psychostimulant Responses

**DOI:** 10.3390/ijms241310909

**Published:** 2023-06-30

**Authors:** Pearl N. Cummins-Beebee, Maggie M. Chvilicek, Adrian Rothenfluh

**Affiliations:** 1Department of Psychiatry, University of Utah, Salt Lake City, UT 84112, USA; 2Molecular Medicine Program, University of Utah, Salt Lake City, UT 84112, USA; 3Neuroscience Graduate Program, University of Utah, Salt Lake City, UT 84112, USA; 4Department of Neurobiology, University of Utah, Salt Lake City, UT 84112, USA; 5Department of Human Genetics, University of Utah, Salt Lake City, UT 84112, USA

**Keywords:** *Drosophila*, substance use disorder, addiction, anhedonia, assays

## Abstract

Addiction is a progressive and complex disease that encompasses a wide range of disorders and symptoms, including substance use disorder (SUD), for which there are few therapeutic treatments. SUD is the uncontrolled and chronic use of substances despite the negative consequences resulting from this use. The progressive nature of addiction is organized into a testable framework, the neurobiological stage-based model, that includes three behavioral stages: (1) binge/intoxication, (2) withdrawal/negative affect, and (3) preoccupation/anticipation. Human studies offer limited opportunities for mechanistic insights into these; therefore, model organisms, like *Drosophila melanogaster*, are necessary for understanding SUD. *Drosophila* is a powerful model organism that displays a variety of SUD-like behaviors consistent with human and mammalian substance use, making flies a great candidate to study mechanisms of behavior. Additionally, there are an abundance of genetic tools like the GAL4/UAS and CRISPR/Cas9 systems that can be used to gain insight into the molecular mechanisms underlying the endophenotypes of the three-stage model. This review uses the three-stage framework and discusses how easily testable endophenotypes have been examined with experiments using *Drosophila*, and it outlines their potential for investigating other endophenotypes.

## 1. Flies Are a Powerful Model to Study Addiction

### 1.1. What Is Addiction?

Addiction is an umbrella term that encompasses a variety of disorders and symptoms, including intense urges to take a drug, spending excessive amounts of time or money to obtain a drug, and withdrawal symptoms in the absence of drug intake. The Diagnostic and Statistical Manual of Mental Disorders (DSM-V) defines addiction as presenting with 2 or more out of 11 possible symptoms, underscoring the heterogeneity of the clinical presentation. Substance use disorder (SUD) costs the human population billions of dollars and contributes significantly to disease and mortality burdens across the world [1]. SUD is characterized as the excessive use of substances and loss of control over this use despite adverse effects that can result in significant impairment of normal functioning. While substance use does not always result in addiction, it is a gateway for behavioral changes that lead to the development of an SUD [2,3,4]. For the purpose of this review, we apply the term addiction specifically to substance addiction and will use it interchangeably with SUD.

Addiction is a progressive disorder that is characterized by the transition from casual substance use to SUD, where an individual physically and psychologically depends on a substance (e.g., alcohol or cocaine). For instance, non-addicted people use substances to feel happy or relaxed, but an addicted person uses drugs to feel “normal” [5,6,7,8,9]. Overall, addiction is a poorly understood disease that is difficult to treat.

### 1.2. The Neurobiological Stage-Based Model Breaks Down Addiction into Behavioral Components

Koob and Le Moal [10] split the progressive nature of addiction into a testable framework called the neurobiological stage-based model. This model suggests that substance-induced neurobiological changes influence three non-mutually exclusive behavioral stages that perpetuate addiction: (1) binge/intoxication, (2) withdrawal/negative affect, and (3) preoccupation/anticipation. This model will be described in detail in section two of this review. 

A hypothetical example illustrates these stages: In the binge/intoxication stage, an alcohol user will drink alcohol and experience positive effects, like euphoria. The person learns to associate these effects with their environment, creating drug cues that remind them of the positive effects, such as drinking while watching TV in the evening. Turning on the TV on another day may thus incite the urge to drink, thereby reinforcing and perpetuating the drinking behavior further. 

As a person drinks chronically, their brain adapts to the high volumes of drinking and the alcohol in their system. First, they develop tolerance, where the positive feelings associated with alcohol are only achieved with increasingly larger amounts of drinking [11,12]. This may then progress to a state of dependence, where alcohol is required to maintain a stable baseline of brain function. In other words, the nervous system adapts to the presence of alcohol, and the presence of alcohol required for normal activity. However, when alcohol is absent from the system, negative consequences like nausea or anxiety arise and alcohol is no longer consumed for its positive effects, but to alleviate aversive withdrawal states (i.e., negative consequences). Furthermore, in the withdrawal/negative affect stage, anhedonia can also arise, manifesting in a reduced reactivity to pleasurable stimuli like food or social interaction. Thus, in general, the effects of the drug have shifted from being positively reinforcing to negative reinforcement, wherein drug use continues in order to ward off negative physiological and emotional consequences of being drug-free. 

If this individual decided to stop alcohol consumption through short-term abstinence (like going to work) or protracted abstinence (stopping drinking altogether), they would transition into the preoccupation/anticipation stage. Here, drug cues that were developed in the binge/intoxication stage (in our example, turning the TV on) will cause an urge to drink, and cravings may lead to the reinstatement of drinking or relapse. In addition to these cues, alcohol-induced executive impairments, such as an increased impulsivity, may also trigger a relapse, thus starting the cycle over again [13]. Overall, this model can be applied to active addiction, where the individual can experience all three stages in a single day, as well as the full addiction journey, where the individual becomes addicted and then chooses to stop overall drug consumption. 

The neurobiological stage-based model is useful for defining and understanding the endophenotypes of addiction—intermediate phenotypes that are measurable components of a larger, more abstract phenotype. These clearly defined phenotypes allow for the development of reproducible assays reflecting the different stages of addiction. For example, we can test hypotheses such as the following: is a specific brain region activated during stage one or is a specific gene associated with a distinct stage-three phenotype? In general, this three-stage model facilitates the definition of specific mechanistic research questions that are clearly circumscribed within a given endophenotype.

One of the reasons that addiction is a complex disease is because multiple forces are known to drive its development. For instance, environmental factors, such as early life trauma [14,15,16,17], and genetic factors, as well as the interaction between environment and genetics, impact the development of addiction. Through analyses of drug use in monozygotic twins, two siblings with exactly the same DNA sequence, the heritability of SUDs has been examined and it has been shown that cocaine addiction is more heritable than addiction to hallucinogens [18]. As a consequence, the risk of developing a cocaine SUD is higher if there is a history of cocaine abuse in the family. Understanding SUD genetics can have positive impacts on addiction outcomes: (1) understanding who is at risk can alter individual behavior to reduce exposure, and (2) understanding genes that contribute to addiction has explanatory power for disease progression and may aid in finding therapeutic interventions preventing progression, or even mitigating current symptoms.

Due to limited opportunities for mechanistic insights from human studies, model organisms are necessary for understanding addiction. Although rodent models have contributed to elucidating the brain regions impacted by drugs and have helped led to the development of some pharmaceutical addiction treatments, there are few high-efficacy interventions for addiction, in large part because the field lacks a thorough understanding of the mechanisms underlying all the stages of addiction. Because addiction is a heritable disease, genetically amendable model organisms are also useful. A genetically engineered rodent can take many months to generate with the tools available, whereas the generation time for the vinegar fly, *Drosophila melanogaster*, is much shorter. Additionally, there are outstanding genetic tools available, making fly genetic manipulations a highly efficient process. Moreover, fly research has a proven record of identifying genes linked to substance-induced behaviors and elucidating the molecular pathways underlying addiction. The following sections will describe this work.

### 1.3. Benefits of Drosophila to Study Addiction

*Drosophila* have been a useful tool in the field of neuroscience for over a hundred years [19]. Their fast generation times (~2 weeks), simple genetic makeup, and low cost make them an appealing model organism to understand genetic and molecular mechanisms involved in the addiction cycle. Flies have many biological features that make them a powerful, translational model for understanding the genetic and molecular mechanisms underlying diseases, including addiction. For instance, the fully sequenced *Drosophila* genome [20] and high percentage (~75%) of human disease-linked *Drosophila* orthologs [21] showcase the genetic conservation and translational opportunities in flies [22]. Many tools exist to genetically manipulate flies, which can be used to validate candidate genes associated with addiction in human Genome-Wide Association Studies (GWAS). For example, the binary GAL4-UAS system [23] uses a promoter that drives the expression of a transcriptional activator (GAL4) in a known spatial and temporal manner and the GAL4-responsive upstream activating sequence (UAS) is linked to effector DNA of interest, for example, inhibitory RNA, to knock down the expression levels of a specific gene. Using the GAL4-UAS system, tissue-specific genetic manipulations can determine how genetic alterations in specific anatomical regions impact behavioral phenotypes. CRISPR-Cas9 [24] is a more recently developed genetic tool that allows for the manipulation of DNA sequences at exact chromosomal locations [25]. Both of these systems allow for fast reverse genetics to quickly test the phenotypes of genes previously studied in addiction. Human GWAS identifies associations between genes and phenotypes, and the two approaches presented above allow for rapid testing of whether a candidate gene affects a specific SUD endophenotype. With these publicly available tools and collection of mutants, we can study nearly every fly gene and how mutations within these genes impact disease phenotypes in flies. 

The three-stage model is a useful tool to begin understanding the molecular mechanisms underlying addiction because it facilitates the design of reproducible assays that mimic the specific stages, or sub-behaviors within those stages. Therefore, one relevant question to ask is if fly behaviors can be mapped onto the neurobiological stage-based model of addiction, a topic that we will discuss below.

### 1.4. Genetic Approaches to Studying Addiction Using Flies 

As mentioned above, one approach to studying the genetic contribution to SUD phenotypes is reverse genetics by validating SUD-associated genes from human GWAS. Through this approach, it was shown that the fly orthologs of the human AUD-associated *AUTS2* and *SLC39A8* genes affect ethanol responses in flies [26,27], thus moving these gene candidates from correlation to causation. Similarly, one can test candidate genes that have been identified in non-SUD-related phenotypes. For example, a number of genes isolated for their function in *Drosophila* circadian rhythms were found to affect flies’ sensitivity to cocaine-induced motor effects [28]. Mouse orthologs of these genes were then shown to affect cocaine-induced behavior in mice as well [29]. Together, these studies illustrate that reverse genetic (gene-to-phenotype) approaches have led to the identification and investigation of genes in flies that also affect mammalian SUD phenotypes.

Historically, one powerful approach in *Drosophila* has been forward genetics, or screening through collections of genetic variants for their phenotype, and then identifying and characterizing the gene responsible for the phenotype. Through this method, RhoGAP18B was found to affect alcohol-induced sedation [30]. From there, other genes upstream and downstream of RhoGAP18B signaling were identified to similarly affect the alcohol sedation phenotype, indicating that these genes are a part of the same molecular pathway, from an upstream cell adhesion molecule to the regulation of the actin cytoskeleton [31,32]. Originating from this work in flies, it was then shown that one member of this pathway, *Rsu1*, has a human ortholog associated with alcohol dependence and drinking, as well as with differential brain activation during a reward anticipation task [32]. This illustrates that unbiased forward genetic approaches screening for AUD-related phenotypes in *Drosophila* can lead to insights into mechanisms that are conserved in human drug responses. This approach has also been replicated for other genes [33,34,35,36].

## 2. Using the Fly to Understand the Literature Gaps in Addiction

The three stages of the neurobiological stage model are (1) binge/intoxication, (2) withdrawal/negative affect, and (3) preoccupation/anticipation [10]. Within these stages, there are measurable behaviors that drive the addiction cycle (Figure 1, Table 1). Although this model was developed based on human behaviors, a lot of work has been conducted in rodents examining the different endophenotypes within these three stages. The question is how these phenotypes map onto the three-stage model in flies.

This review will primarily discuss literature on alcohol and psychostimulants in flies because many other drugs of abuse have not been used extensively in fly addiction. Still, flies have been studied with regard to sensitivity to opiates [37]. Opiates are challenging to study in flies because they do not have obvious opioid receptors; however, evolutionary approaches are being utilized to address this challenge [37]. Additionally, flies have been investigated for nicotine sensitivity [38]. Again, not much is known about nicotine addiction, partly because nicotine exposure can induce seizures.

**Figure 1 ijms-24-10909-f001:**
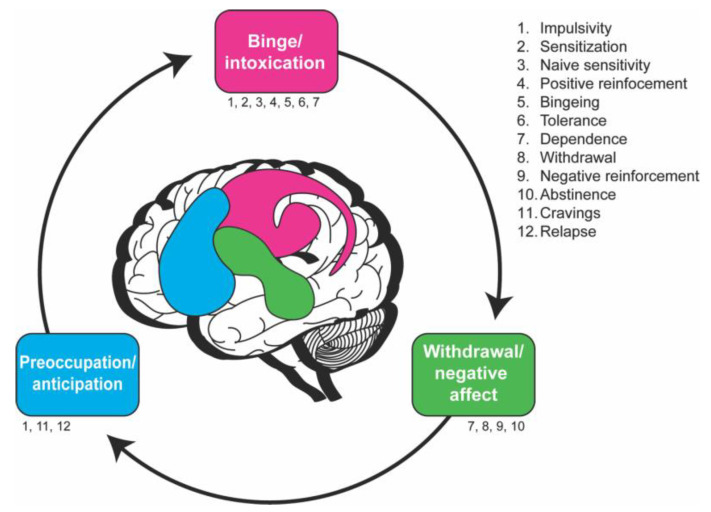
The cycle of addiction as explained with the neurobiological stage-based model in humans is characterized by proposed endophenotypes within each stage. The ventral tegmental area, substantia nigra, dorsal striatum, ventral and dorsal globus pallidus, and thalamus (pink) are involved in the binge/intoxication stage, while the extended amygdala and hypothalamus (green) are involved in the withdrawal/negative affect stage and the prefrontal cortex, hippocampus, insula, and basolateral amygdala (blue) are involved in the preoccupation/anticipation stage. The list (right) and text below each stage indicate the proposed behavioral characteristics of that stage. These endophenotypes are not necessarily stage-specific and some occur in multiple stages. This schematic is adapted from the Substance Abuse and Mental Health Services Administration [39].

**Table 1 ijms-24-10909-t001:** The neurobiological stage-based model with endophenotypes and fly assays.

Endophenotypes	Neurobiological Stage(s) of Addiction	*Drosophila* Assays	References
Sensitization/tolerance	Binge/Intoxication, Withdrawal/Negative affect	Booze-o-mat (E)	[40,41]
		(Drug exposure through forced vaporization; determine locomotion).	
		Maples assay (E)	[42]
		(Drug exposure through forced vaporization; determine loss of righting).	
		Inebriometer (E)	[36,43]
		(Drug exposure through forced vaporization; determine loss of postural control).	
		FlyBong (C, M)	[44,45]
		(Drug exposure through vaporization and forced administration).	
		CApillary FEeder assay (CAFÉ; E)	[46]
		(Drug exposure through voluntary 2-choice food consumption; determine volume consumed and preference).	
		FlyCafe (M)	[43]
		(Drug exposure through voluntary 2-choice food consumption; determine preference and effect on locomotion).	
		Fly group activity monitor assay (FlyGrAM; E)	[47,48]
		(Drug exposure through vaporization and forced administration. Locomotor activity assay).	
		Drosophila activity monitor (DAM; M)	[45]
		(Locomotor activity assay of variably drug-exposed flies).	
		DAM5M (M)	[45]
		(Locomotor activity assay of variably drug-exposed flies).	
Naïve sensitivity	Binge/Intoxication	Inebriometer (E)	[49,50]
		Booze-o-mat (E)	[30,40]
		Maples assay (E)	[42]
		FlyGrAM (E)	[47,48]
Positive reinforcement	Binge/Intoxication	Fly liquid–food interaction counter (FLIC)	[51]
		(Drug exposure through voluntary 2-choice food consumption; determine food interaction time and preference).	
		Fluorometric Reading Assay of Preference Primed by Ethanol (FRAPPE)	[52]
		(Drug exposure through voluntary 2-choice food consumption; determine amount ingested and preference).	
		CAFÉ	[52,53]
		FlyCafe (M)	[45]
		Proboscis extension reflex (PER) assay (E)	[54]
		(Consummatory reflex of variably drug-exposed flies).	
		T-maze (E)	[55]
		(Olfactory choice assay, or variably drug-exposed flies).	
		DAM (M)	[45]
Bingeing	Binge/Intoxication	Not yet developed	
Dependence	Binge/Intoxication, Withdrawal/Negative affect	Drug feeding in combination with learning and memory assay (E)	[56]
Withdrawal	Withdrawal/Negative affect	Drug feeding in combination with learning and memory assay (E)	[56]
Negative reinforcement	Withdrawal/Negative affect	Not yet developed	
Abstinence	Withdrawal/Negative affect	Drug feeding in combination with learning and memory assay (E)	[56]
Cravings	Preoccupation/Anticipation	Not yet developed	
Impulsivity	Binge/Intoxication, Preoccupation/Anticipation	Not yet developed	
Relapse	Preoccupation/Anticipation	Not yet developed	

For each endophenotype, published *Drosophila* assays are shown along with the neurobiological stage(s) relevant for that behavior. This table showcases studies that have used these assays for exposure to a specific drug; however, these assays are not limited to the indicated drugs and can possibly be adapted to study other substances. E = ethanol, M = methamphetamine, C = cocaine.

### 2.1. Stage 1—Binge/Intoxication

The binge/intoxication stage consists of substance-induced positive reinforcement through stimulation of the brain’s reward circuit. Classical addiction research has mainly focused on this stage because addiction has historically been conceptualized as an increase in drug taking. However, this historical classification has limited much of our knowledge about addiction and its mechanisms primarily to the binge/intoxication stage, thus omitting other important endophenotypes seen in the other two stages of the addiction cycle.

Stage one has been extensively studied in rodents using self-administration assays. A fundamental component of self-administration assays in addiction research is the ability of the organism to have control over its drug consumption, consistent with human addiction. The mesocorticostriatal dopamine (DA) systems are the key contributors to stage one and are largely linked to the rewarding effects of drugs (e.g., euphoric high) [57]. These rewarding drug properties are key factors in what makes drugs addictive. Though the mechanisms of action are different for alcohol and psychostimulants, they all converge on the human reward system, where many neurotransmitters are conserved with flies [58,59,60]. Cocaine blocks DA reuptake into the presynaptic terminal, leaving large amounts of DA in the synapse. This excess DA continuously activates DA postsynaptic neurons, leading to a sense of euphoria and behavioral reinforcement [61]. An amphetamine additionally increases the output of DA into the synapse from the presynaptic terminal, thereby increasing postsynaptic neuronal excitation [62,63,64]. Thus, both of these psychostimulants elevate the amount of DA in the synapse. Human imaging studies show an increase in DA after intoxicating administration of alcohol [65,66]. In humans and flies [55], DA is a key player in the reward system [67] and mediates reinforcing properties of drugs of abuse [68]; together, this showcases the shared mechanisms in flies and humans that perpetuate addiction.

#### 2.1.1. Behaviors Underlying the Binge/Intoxication Stage

Locomotion is often altered after drug administration, is conserved across species, and is a common method for measuring behaviors within the binge/intoxication stage. The biphasic alcohol response, when the nervous system experiences a phase of stimulation and then a phase of depression after alcohol administration, is a prime example of how locomotion can be a readout of the CNS effects of drugs. For example, CNS-stimulating doses of alcohol in flies and humans cause increased locomotor activity. As alcohol absorption continues, CNS depression causes decreased locomotor activity and eventually sedation [58]. In other words, low to moderate doses of alcohol in flies and humans causes one effect (hyperactivity) while larger doses of alcohol cause a different effect (sedation). The locomotor-activating phase of alcohol has been studied in flies [40] and it requires the activity of DAergic neurons projecting to the ellipsoid body pre-motor center [69]. Additionally, flies and rodents under the influence of psychostimulants, like cocaine or a methamphetamine, will experience enhanced acute locomotor activity [59,70,71]. Locomotor activity is a useful output metric of initial drug effects that can help us measure if and when the following endophenotypes occur.

##### Sensitization

Flies and humans can develop drug sensitization after a brief exposure to a drug, resulting in stronger drug-induced effects afterwards [71]. Sensitization is most well-studied in the context of locomotion. Psychostimulant exposure sensitizes rodents to the locomotor activating effects during a subsequent exposure, i.e., the same dose causes increased locomotion after the second exposure. Similarly, flies exposed to a vaporized methamphetamine display increased locomotor activity after the initial exposure and double the amount of locomotor activity after a second exposure [45,59]. For locomotor-activating doses of ethanol, sensitization has not been examined for locomotion, but courtship initiation is sensitized with repeat ethanol exposures, a response that again depends on DAergic activity [72]. Similarly, repeated exposure to psychostimulants results in progressive sensitization to the effects of these drugs in mammals and humans, including sensitization of the DA system [73,74,75].

##### Naïve Sensitivity

In flies, locomotor activity can be used to measure drug sensitivity—how strongly the drug activates locomotion or how effectively the drug intoxicates the organism, leading to the loss of movement and righting ability (when keeled over). Flies and humans can have increased naïve sensitivity to drugs, where they initially experience stronger and faster effects of the drug. A prime example of a naive sensitivity phenotype is observed in humans with an alcohol dehydrogenase deficiency, a genetic condition where there is a mutation in the ALDH2 gene. ALDH2 encodes the aldehyde dehydrogenase enzyme that breaks down acetaldehyde, the noxious metabolite of alcohol. A mutation in ALDH2 results in a reduced breakdown of acetaldehyde and causes carriers to experience more severe negative side effects of alcohol, like headaches, palpitations, and dizziness [76,77,78]. Carriers of these alleles are very unlikely to become alcoholics, and there is a general correlation between the risk of developing AUD and resistance to the naïve intoxicating effects of alcohol [79]. People who experience naïve sensitivity to a drug consume less of the substance since they experience the negative effects of intoxication more quickly; therefore, they are less likely to develop an addiction. Because of this correlation, and because sensitivity to the naïve intoxicating/sedating effects of ethanol is easily determined, many investigations in *Drosophila* use sensitivity assays to determine a gene’s involvement in ethanol responses. However, while aforementioned fly *Rsu1* mutants are resistant to ethanol-induced sedation and do show an increased preference for ethanol (as predicted from the human correlation) [32], this relationship is not universally seen [33,80], and changes in naïve drug responses should be viewed as starting points in understanding a gene’s effect on drug self-administration. Accordingly, many mutations affecting naïve sensitivity also affect self-administration, underscoring the value of this metric for guiding investigations.

Measuring naïve sensitivity is not limited to alcohol, and flies can also show differences in naïve sensitivity to psychostimulants. Using two different behavioral assays, the crackometer to determine cocaine-induced locomotor impairment [81], and a locomotion speed tracking system [82,83], *Lmo* was shown to be involved in cocaine-induced sensitivity to motor impairment. Tsai and colleagues [84] found that the loss of *Lmo* function in pacemaker cells that regulate circadian locomotor activity in flies [85] results in an increased sensitivity to cocaine-induced motor impairment. When this gene was silenced, flies became impaired by cocaine more quickly. Subsequent experiments showed that global downregulation of the *Lmo* mammalian homolog, *Lmo4*, increased sensitivity to locomotor activation after cocaine administration [86]. Moreover, this effect was recapitulated when *Lmo4* was knocked down specifically in the nucleus accumbens, a mammalian brain region strongly linked to addiction and a major target site of DAergic input [86]. The similar phenotypes of homologous genes showcases that molecular mechanisms of sensitivity are likely conserved across invertebrates and vertebrates.

##### Positive Reinforcement

Positive reinforcement occurs when a stimulus reinforces an approach or consummatory behavior. In drug addiction, activation of the reward system reinforces the behavior of ingesting a drug. Positive reinforcement occurs via the reward system, which is primarily composed of DA neurons. In flies, the DAergic cluster of protocerebral anterior medial (PAM) neurons projects to the mushroom body (MB), the learning center, and are linked to positive reinforcement learning [87,88,89,90,91,92,93,94,95]. Although flies naturally avoid many substances of abuse, they can learn to prefer those substances over repeated exposures [52,53]. For example, when flies are initially exposed to alcohol or a methamphetamine, they avoid it, but then begin to choose drug-containing food over a non-drug source within 2 days [45,52,53]. This change in self-administration behavior requires the MB; a loss of *Rsu1* in the MB specifically led to abolished learning of alcohol preference, whereas MB-specific manipulations of the *Rsu1* signaling pathway in the opposite direction caused a faster acquisition of ethanol preference [96]. Moreover, the human ortholog, *Rsu1*, is associated with differential activation in the nucleus accumbens during a reward task [32], thus highlighting the mechanistic conservation of *Rsu1*′s involvement in alcohol preference across species. It is also worth noting here that *Rsu1* was initially isolated as a fly mutant resistant to ethanol-induced sedation, reiterating that easily scored phenotypes, like sedation, can be valuable entry points to understanding more complex behaviors, such as voluntary self-administration. Furthermore, it illustrates that even in flies, a gene can have multiple ethanol-related phenotypes.

The molecular mechanisms underlying the learned preference of psychostimulants in flies are largely unknown, but flies require a normal DAD1 receptor (Dop1R1) for a preferential methamphetamine [97], and normal DA transporter activity for amphetamine self-administration [98]. Currently, there is no known assay for preferential cocaine self-administration; however, we are beginning to understand why this might be. One reason flies avoid cocaine is because of its bitter taste that is detected through peripheral mechanisms in their legs [99]. Moreover, when bitter gustatory neurons are silenced in flies, they avoid cocaine less. Although cocaine preference has not been described yet, this work begins to disentangle the reasons behind why a fly model of cocaine preference remains to be established. However, once a fly model for cocaine preference exists, it will be possible to test the numerous genes that have been associated with human cocaine use for their causal role in cocaine self-administration.

##### Bingeing

Bingeing is the ingestion of a large amount of a drug in a short time period, followed by a period of abstinence. The drug dosage defined as bingeing varies depending on the drug. For instance, alcohol consumption in humans is considered a binge when it leads to a blood alcohol percentage of 0.08% or more [100], whereas cocaine and methamphetamine ingestion is considered a binge when there is a rapid escalation to a high dose over 3 days [101] or 4 days a week [102,103,104,105,106], respectively.

Binge drinking is largely thought to impact the frontal lobe [107,108] as well as the reward pathways and other brain regions associated with addiction. Although alcohol is known to impact brain function at low doses, binge drinking is thought to impact the frontal lobe more severely with more permanent effects than light drinking [107]. More specifically, binge drinking is associated with a decreased frontal lobe volume in humans [107]. Additionally, alcohol bingeing is thought to modulate neuronal excitability through brain region-specific changes in the neuronal firing of major neurotransmitter systems (like glutamatergic and GABAergic neurons) [109].

There is no model of drug bingeing in flies; however, it should be possible to measure both features necessary to classify bingeing: (1) the duration of ingestion and (2) the amount of drug ingested. Recent assays, like the fly liquid–food interaction counter (FLIC; Figure 2), have the potential to measure binge-like behaviors in the fly. The FLIC can specifically measure the duration of food interaction with a fly’s proboscis—a mouth-like structure used to consume food—which is proportional to the amount of food ingested [51]. Because the FLIC can be used longitudinally, over days, one could determine the development of interaction preference, as well as the potential increase in drinking bouts, akin to bingeing, as flies learn to prefer alcohol. Additionally, one could pre-expose flies to alcohol chronically, and ask whether that leads to drinking binges compared to unexposed flies.

### 2.2. Stage 2—Withdrawal/Negative Affect

The withdrawal/negative affect stage occurs upon drug-induced neurochemical changes in the brain that result in a change in the responses to drugs. During the binge/intoxication stage, substance use occurs to elicit positive feelings, but in the withdrawal/negative affect stage, the motivation to take a substance is mainly to prevent withdrawal and other negative symptoms [110]. This stage is thought to involve parallel processes, a decrease in reward function and an increase in stress function, which both contribute to a shift in motivation from reward-seeking to withdrawal avoidance. After chronic drug exposure, neuroadaptations occur in the brain to counteract the effects of the drug. These adaptations contribute to a decrease in reward function [111]. For instance, DAergic responses to the drug or other normally rewarding stimuli decrease in both rodents [112] and humans during withdrawal. Other neuroadaptations also involve GABA [113] and glutamatergic transmission [114].

Additional drug-induced neuroadaptations can occur in non-reward circuits, which are between-systems neuroadaptations that can increase stress reactions [111]. These antireward circuits are recruited during stage two of the neurobiological stage-based model and are involved in the shift in motivation to use a drug for positive effects to the motivation to avoid aversive psychological and physiological symptoms, including withdrawal symptoms [115]. A variety of molecules (e.g., corticotropin-releasing factor, dynorphin) that are involved in brain stress systems play a role in negative affective behaviors (e.g., irritability) and are thought to be recruited during the development of tolerance, or the decrease in drug-induced responses after repeated drug administration [116]. Many molecular mechanisms underlying motivation and the between- and within-system circuitry are still unclear, but some of the brain areas and neurotransmitters involved in the between-system circuit are known. For example, the hypothalamic–pituitary–adrenal axis and the extended amygdala are thought to play a role in stress and anxiety, a common withdrawal symptom in addiction [117]. Additionally, the activation of the norepinephrine stress system is associated with negative consequences in stage two [110]. However, there is little information known about the molecular mechanisms in between-system circuits that are affected by overactive reward circuits.

#### 2.2.1. Behaviors Underlying the Withdrawal/Negative Affect Stage

##### Tolerance

Tolerance is a decrease in responsiveness (or sensitivity) after repeated drug exposure and is most easily measured in flies through locomotor activity and a loss of righting, though other measures such as alcohol-induced hypothermia also show tolerance with repeat exposure. Within the domain of tolerance, there are two subcategories—metabolic tolerance and functional tolerance. Metabolic tolerance reflects an increasingly rapid elimination of a drug from the body through enzymatic action. Functional tolerance involves changes at the neuronal level that influence behavior when a substance is introduced to the nervous system in the absence of changes in the pharmacokinetics of the drug [118]. These neuroadaptations require the person or the fly to administer increasing amounts of the substance to achieve the same behavioral state. *Drosophila* show functional tolerance to ethanol-induced sedation as early as 2 hours after their first high-dose alcohol exposure [118]. Numerous genes have been implicated in ethanol-induced tolerance in *Drosophila* [119], and a link between the ability to develop tolerance and alcohol preference has been suggested [80].

##### Dependence

With continued repeat exposure, tolerance results in dependence, where drug consumption is needed to function normally. Dependence can be broken up into two categories: physical and cognitive state dependence. Generally, not much information is known about the molecular mechanisms that underlie the development of either type of dependence.

Physical dependence occurs when the chronic use of drugs induces neuroadaptations to bring the brain back to homeostasis. In other words, the brain physically depends on the drug to function normally. Without the drug, humans, rodents, and flies can experience physical withdrawal symptoms like alcohol withdrawal seizures. Physical dependence has not been directly described in flies; however, it is often ascertained by the occurrence of withdrawal symptoms upon cessation of drug exposure as a proxy for physical dependence. For ethanol, withdrawal symptoms have been described (see below) [120].

Cognitive state dependence occurs when a substance is needed to perform normally on cognitive tasks, like memory tests. In the absence of ethanol, chronically ethanol-exposed *Drosophila* larvae showed reduced learning [56]. Additionally, the same larvae showed normal learning when they were given ethanol an hour later [56]. This indicates that the chronically ethanol-exposed larvae were dependent on ethanol to normally perform the learning task [56]. Thus, even so-called higher-order processes, like state-dependent learning, can be modeled in *Drosophila*.

##### Withdrawal

When a drug user is in a state of dependence but does not have that drug in their system, they can experience aversive physical and psychological symptoms known as withdrawal. Humans and flies develop alcohol withdrawal syndrome, where neurons become hyperexcited and induce seizures [120,121,122]. This syndrome acts upon the major excitatory and inhibitory neurotransmitters, glutamate and GABA. Chronic alcohol exposure results in a decrease in GABA neuronal function, thus decreasing the inhibition of excitatory glutamatergic neurons and making the brain more excitable. Bayard and colleagues have a more in-depth explanation of alcohol withdrawal syndrome [121]. To investigate withdrawal symptoms in Drosophila, Ghezzi and colleagues [120] determined fly seizure thresholds, i.e., how much current is needed to induce a seizure. They found that after ethanol exposure, once alcohol has cleared the system, flies showed a reduced seizure threshold, meaning that ethanol-adapted flies are more susceptible to seizures [120]. Additionally, this decrease in seizure thresholds was found to be dependent on the expression of a specific gene, *slo*, encoding a potassium channel involved in the resting membrane potential, and with it, the likelihood of neuronal firing [120].

In humans, both alcohol and cocaine withdrawal has been associated with a decreased DA release. Additionally, it has recently been discovered in rodents that DA can alter gene expression in the ventral tegmental area, a brain region implicated in the reward circuit and addiction, during withdrawal [123]. Both pieces of information indicate altered DA function during withdrawal.

##### Negative Reinforcement

Negative reinforcement leads to the increasing likelihood of a particular behavior to avoid or prevent a negative outcome. In addiction, negative reinforcement is the use of drugs to avoid or alleviate withdrawal symptoms like negative affect. Negative affect is challenging to study in animal models because ‘affect’ refers to the emotional well-being of the organism, a state that can only be communicated through language. However, we can study the behavioral consequences of negative affect, including anhedonic-like behaviors, which will be discussed below.

Anhedonia is the reduced reactivity to pleasurable stimuli and reflects a dysfunction of the reward system. In the context of addiction, drugs cause neuroadaptations that result in anhedonia, a phenomenon known as hedonic tolerance. Hedonic tolerance is measured using elevations in a reward threshold, i.e., how large a reward has to be in order to be appetitive/reinforcing, during a state of withdrawal. Anhedonic-like behaviors are behavioral outputs that can be used to detect the presence of hedonic tolerance [124,125]. For a person chronically using a drug, the reward system is repeatedly overactivated, and it is thought that neurobiological changes occur in the reward system that increase the reward threshold [10]. This means that due to hedonic tolerance, other pleasurable stimuli, such as food and social interaction, become less rewarding, and continued drug use in animal models as well as humans will not be as rewarding as the initial experience [116].

Many aspects of the neuronal circuitry and molecular mechanisms underlying the withdrawal/negative affect stage in humans and flies remain unclear, particularly those relating to hedonic tolerance. Furthermore, it is unclear whether mechanisms and genes involved in tolerance to the intoxicating/sedating effects of a drug, mostly measured using motor impairment, are also involved in tolerance of the hedonic system. Given that numerous genes in *Drosophila* are known to affect tolerance to sedation, this would be a testable hypothesis—if flies do actually develop hedonic tolerance with repeat drug exposure. It remains to be determined whether assays used to measure anhedonia/hedonic tolerance in rodents, such as sucrose consumption or preference, can be used in flies to investigate drug-induced hedonic tolerance.

### 2.3. Stage 3—Preoccupation/Anticipation

The preoccupation/anticipation stage is characterized by the anticipation and reinstatement of drug-seeking behavior after acute or protracted abstinence. It is thought that neuroadaptations in the prefrontal cortex and projections connecting the reward system to the frontal lobe play a role in incentive salience, or the motivation to attain a reward [126,127]. The behaviors involved in this stage include abstinence, cravings, impulsivity, and relapse (drug-, cue-, or stress-induced; Figure 1).

#### 2.3.1. Behaviors Underlying the Preoccupation/Anticipation Stage

##### Abstinence

Abstinence refers to a self-imposed or forced period of time when the drug is not being consumed. Often, this behavior occurs when a drug user stops using the drug during addiction treatment; however, abstinence also occurs when drugs are simply not available to use. The majority of abstinence models in animals involve forced abstinence through the extinction of operant behavior (withholding the drug) [128]. A comprehensive explanation of these models can be found in Peck [128]. Abstinence has not been investigated extensively in *Drosophila*. However, after gradually acquiring a preference for alcohol consumption and being subjected to a brief period (1–3 days) of forced abstinence, flies retain the preference for alcohol consumption, and they do not have to re-learn the preference [53]. This indicates that the preference for alcohol is a lasting memory [52] and other behaviors in this stage might be amenable to experimental investigation.

##### Cravings

Drug cravings are a motivational state that influences drug-seeking behavior through environmental cues made in stage one and is a key feature in the DSM-V classification of substance use disorder [129]. Currently, in animal models, cravings can only be measured indirectly through cue-induced paradigms. It is thought that cravings are, in part, induced by stress through the activation of the hypothalamo–pituitary–adrenal (HPA) axis and the prefrontal cortex. For instance, many drug users seem to have their addiction under control until they face a stressful event [130]. Drug cravings are also likely to involve stress molecules like corticotropin-releasing factor (CRF) and cortisol.

Additionally, drug cravings are linked to adaptations in DAergic and glutamatergic neurons [131,132]. PET imaging of the human dorsal striatum shows correlations between DA receptor binding and cocaine cravings [133,134] as well as alcohol cravings [135], [136]. In rodents, endogenous DA signaling is necessary for drug-induced cravings [137]. More research is needed to further understand the molecular mechanisms underlying this behavior. Furthermore, while flies show the extinction of memories [138], it is unclear whether that might also hold for drug-associated cues [55], and if the drug-induced cue approach could be reinstated by stress or a re-exposure to the drug.

##### Impulsivity

Impulsive behaviors occur to promote gratification [139] and precipitate the reinstatement of drug-seeking behavior after an individual is presented with environmental cues, drug cues, or stress. Note that impulsive behavior occurs at the beginning of the binge/intoxication stage and in the preoccupation/anticipation stage. Generally, not much information is known about the molecular mechanisms underlying this behavior in humans and flies; however, the human gene, *GPM6B*, has recently been associated with impulsive decision-making [140]. The link between *GPM6B* and impulsivity opens a door of opportunity for researchers to begin understanding the molecular mechanisms of impulsivity, especially because *GPM6B* has an ortholog in the fly, *M6*. One can investigate the mechanism of *M6* by manipulating this gene in flies and observing how *M6*-manipulated flies behave in impulsivity assays, like the fly liquid–food electroshock assay (FLEA) [141]. The FLEA is a two-choice assay similar to the FLIC (Figure 2) that incorporates an electric shock as a variable. The FLEA is similar to mammalian impulsivity assays, like the Modified Vogel’s conflict choice [142], where rats receive an electric shock upon drinking from a spout.

##### Relapse

Impulsive behavior after abstinence can result in relapse, which is the reinstatement of drug abuse. Three types of reinstatement, drug-induced (caused by contact with the drug of abuse), cue-induced (caused by contact with a stimulus associated with the drug of abuse), and stress-induced (caused by the presence of a stressful stimulus), can lead to relapse [143]. In rodent models, drug- and cue-induced reinstatement of drug-seeking behavior involves glutamatergic projections from the prelimbic prefrontal cortex to the reward system and is modulated by frontal lobe DA receptors [144,145,146], though other mechanisms likely play a role as well. Additionally, stress-induced reinstatement of drug-seeking behavior likely involves glutamatergic responses and CRF systems after protracted abstinence [147,148,149]. More specifically, Zhao and colleagues [149] found that the activation of a subset of glutamate receptors in the hippocampus and amygdala reduces stress-induced alcohol-seeking behavior in rats. It is also hypothesized that these glutamatergic projections are involved in mediating craving-like responses. Overall, the mechanisms underlying drug-, cue-, and stress-induced drug-seeking behavior need more research so that scientists can better understand relapse, a driving force of the addiction cycle.

## 3. Conclusions

The neurobiological stage-based model provides an important framework to understand the addiction cycle. Though there is a lot of research on addiction, most research focuses on the binge/intoxication stage, and little information exists about the biological mechanisms of the withdrawal/negative affect and preoccupation/anticipation stages. However, the circuits and mechanisms that affect the different stages may overlap, and insights from one stage may lead to insights in other stages too. To take another example from flies, the fly *dPsd* gene was isolated due to its sensitivity phenotype in ethanol-induced sedation. These mutants also develop less tolerance, and impact alcohol self-administration [33]. Variants in the human *PSD3* ortholog are associated with the alcohol drinking frequency and dependence. In addition, they are also associated with the differential activation of the prefrontal cortex during a go/no-go task, which tests executive control and impulsive action [33]. The *dPsd/PSD3* genes thus affect behaviors in stages one and two in flies, and stages two and three in humans. The findings further illustrate the genetic conservation between flies and mammals and how flies can serve as a starting point to gain insights into human addiction. Notably, *PSD3* has also recently been associated with cocaine use [150], reiterating that many insights into the mechanisms of addiction apply to multiple drugs.

*Drosophila* is a powerful model organism that has been used for many years to understand the mechanisms of disease, including fundamental aspects of addiction. However, there are limitations to using flies in research. For instance, some of the neurotransmitters in mammals (e.g., norepinephrine) are not the same in flies, though they do have fly orthologs (e.g., octopamine). Additionally, flies do not have all the cell types that mammalian nervous systems have, and their brains are obviously less complex. Lastly, some of the more complicated behavioral assays that are possible in higher mammals, such as rats [151], may be difficult to ever model in *Drosophila*. Though flies do have limitations, their experimental amenability and economy of scale are obvious benefits. For instance, flies are an efficient genetic tool that can identify genes and molecular pathways underlying key behaviors that reinforce the addiction cycle. Additionally, fly assays can often be created using inexpensive equipment available in most labs, and can be tested and validated much more quickly than mammalian behavioral assays. Understanding the genes involved in susceptibility to addiction can be useful in both preventative care for individuals who are non-addicts as well as therapeutic care for individuals who are currently caught in the repetitive cycle of addiction.

## Figures and Tables

**Figure 2 ijms-24-10909-f002:**
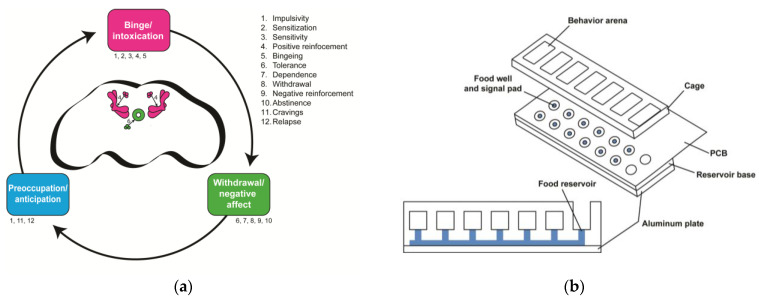
*Drosophila* offer opportunities for insights into the neurobiological stage-based model of addiction. (**a**) How the proposed three-stage model of addiction maps onto the fly brain. Each stage is labeled with numbers that correspond to the endophenotypes listed on the right. These endophenotypes are not necessarily stage-specific and some occur in multiple stages. The fly brain schematic shows the mushroom body and the PAM neuronal DA cluster (pink), which are involved in the binge/intoxication stage, and the ellipsoid body and PPM neuronal DA cluster (green), which are involved in the withdrawal/negative affect stage. No brain regions are known to be involved in the preoccupation/anticipation stage. (**b**) The Fly Liquid–Food Interaction Counter (FLIC) assay can be used to study endophenotypes in the binge/intoxication stage (e.g., positive reinforcement and bingeing) in flies. The assay is assembled with an aluminum plate at the base, the reservoir base, and the printed circuit board (PCB). The reservoir base is where the solutions are stored, and the PCB contains the electrical circuitry needed to measure the time flies spend interacting with the solution. Flies are transferred to the behavior arenas where they have the choice between solutions in two food reservoir wells. The food wells are surrounded by a signal pad where flies close an electrical circuit upon interacting with the solution. Software measures the amount of interaction time that flies have with the solution. This schematic is adapted from Ro et al., 2014 [51].

## Data Availability

Not applicable.

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
