# Peer review of "The Stage-Based Model of Addiction—Using Drosophila to Investigate Alcohol and Psychostimulant Responses"

_ijms, 2023, doi:10.3390/ijms241310909_

Round 1

Reviewer 1 Report

The review article entitled “The stage-based model of addiction – using Drosophila to 2 investigate alcohol and psychostimulant responses” by Pearl et al for publication in IJMS. Overall, the article looks great to me as it opens the door for study using Drosophila as a model organism for the study of addiction-based studies.

Authors should discuss different Drosophila based assay in more details used for stage-based alcohol addiction or can be used for other drug’s addiction-based study in Drosophila.

Here authors summarize the different neurological stage models of alcohol addiction in humans and Drosophila. But the authors didn’t discuss much about the other drugs-based additions and whether we can use Drosophila based neurobiological methods for other drugs addiction study. 

Authors should also discuss the limitation of the Drosophila model for addition-based study.

Author Response

We thank the reviewers for their constructive comments on our manuscript.

We have edited the original submission with track changes visible.

Specifically, we have addressed:

Reviewer 1, comment 1: Authors should discuss different Drosophila-based assay in more details.

We have added short descriptions to the Table. For space concerns, we have not added too much detail on the assays, but each assay is presented next to its citation, for easy lookup.

Rev.1, cmt.2: …authors didn’t discuss much about other drug-based addictions.

We have clarified our focus on alcohol and psychostimulants in this review, but also briefly refer to other drugs and how they have been studied in flies. See line 263+ in revised manuscript.

Rev.1, cmt.3: authors should also discuss the limitations of Drosophila.

We have done so in lines 659+.

Reviewer 2 Report

The review article titled "The stage-based model of addiction - using Drosophila to investigate alcohol and psychostimulant responses" is well-organized and extensive. The authors have discussed about the substance use disorder (SUD) and they discuss how Drosophila melanogaster, are necessary for understanding SUD. The article is very interesting and needs to be published.

Here are my comments -

1)      A more concise presentation of the Introduction section is required.

2)      In the section 2.2.1- line 413, the authors mentioned numerous genes have been involved in ethanol-induced tolerance in Drosophila. Can they mention the list of genes?

Minor comment -

Line 316 – There is a space between DA D1 receptor. Please correct it.

Author Response

We thank the reviewers for their constructive comments on our manuscript.

We have edited the original submission with track changes visible.

Specifically, we have addressed:

Rev.2, cmt.1: A more concise presentation of the Introduction section is required.

We have cut the introduction in a number of places.

Rev.2, cmt.2: line 413, the authors mentioned numerous genes have been involved in ethanol-induced tolerance in Drosophila. Can they mention the list of genes?

Already presenting a large table, we felt that a table of genes would be too much in this review. However, at the spot pointed out (now line 498), we have added a reference with a gene table.

Rev.2, cmt.3: line 316 typo.

Corrected, thanks! (now, line 400)